# The benefits of action to reduce household air pollution (BAR-HAP) model: A new decision support tool

Ipsita Das[1], Jessica J. Lewis[2], Ramona Ludolph[2], Melanie Bertram[3], Heather Adair-Rohani[2], Marc Jeuland[1,4]*

**1** Sanford School of Public Policy, Duke University, Durham, NC, United States of America, **2** Department of Environment, Climate Change and Health, World Health Organization, Geneva, Switzerland, **3** Department of Health Systems Governance and Financing, World Health Organization, Geneva, Switzerland, **4** Duke Global Health Institute, Duke University, Durham, NC, United States of America

* marc.jeuland@duke.edu

## Abstract

Cooking with polluting and inefficient fuels and technologies is responsible for a large set of global harms, ranging from health and time losses among the billions of people who are energy poor, to environmental degradation at a regional and global scale. This paper presents a new decision-support model–the BAR-HAP Tool–that is aimed at guiding planning of policy interventions to accelerate transitions towards cleaner cooking fuels and technologies. The conceptual model behind BAR-HAP lies in a framework of costs and benefits that is holistic and comprehensive, allows consideration of multiple policy interventions (subsidies, financing, bans, and behavior change communication), and realistically accounts for partial adoption and use of improved cooking technology. It incorporates evidence from recent efforts to characterize the relevant set of parameters that determine those costs and benefits, including those related to intervention effectiveness. Practical aspects of the tool were modified based on feedback from a pilot testing workshop with multisectoral users in Nepal. To demonstrate the functionality of the BAR-HAP tool, we present illustrative calculations related to several cooking transitions in the context of Nepal. In accounting for the multifaceted nature of the issue of household air pollution, the BAR-HAP model is expected to facilitate cross-sector dialogue and problem-solving to address this major health, environment and development challenge.

## 1. Introduction

Exposure to air pollution, especially fine particulate matter (i.e., $PM_{2.5}$) is associated with increased hospitalization, disability and death from a wide range of illnesses [1]. Many people imagine that air pollution exposure is primarily experienced outdoors, but household sources such as cooking, lighting and heating are also critical risk factors for these illnesses, especially in low- and middle-income countries (LMICs). Household air pollution (HAP) from cooking with polluting fuels (e.g., firewood, charcoal, coal, animal dung and crop residue) and inefficient technologies (such as rudimentary stoves, space heater and lamps) is estimated to cause

WHO logo is not permitted. This notice should be preserved along with the article's original URL.

**Data Availability Statement:** The tool and accompanying data are in the process of being posted publicly on the WHO website, and will be available by the time the article is published. In the revised manuscript, we have provided the web link to the published BAR-HAP Excel tool and the corresponding user's manual that includes all model equations. Readers can now access the data sources cited in the manual or access those used in the model itself. https://www.who.int/airpollution/household/interventions/chest-module3-BAR-HAP-tool/en/#:~:text=The%20WHO%20Benefits%20of%20Action,cooking%2Drelated%20household%20air%20pollution.

**Funding:** This study was commissioned and paid for by the World Health Organization (WHO). Copyright in the original work on which this article is based belongs to WHO. The authors have been given permission to publish this article. The authors alone are responsible for the views expressed in this article and they do not necessarily represent the views, decisions or policies of the institutions with which they are affiliated.

**Competing interests:** The authors have declared that no competing interests exist.

approximately 3.8 million deaths per year, with the main health conditions responsible for these deaths being acute lower respiratory infections or ALRI (27%), ischemic heart disease or IHD (27%), chronic obstructive pulmonary disease or COPD (20%), stroke (18%) and lung cancer (8%) [2]. In light of these significant health risks, WHO issued the "Guidelines for indoor air quality: household fuel combustion" (hereafter the WHO Guidelines, available at: https://www.who.int/airpollution/guidelines/household-fuel-combustion/en/), which provide normative recommendations on the emission rates for fuels and technologies that ensure minimal health risk, policies for the transition from current practices to widespread adoption of clean fuels and technologies, and recommendations to discourage household use of kerosene and eliminate the use of unprocessed coal [3].

The WHO Guidelines notwithstanding, approximately 37% of the world's population continue to lack access to any clean cooking fuel and technology; most of these reside in Sub-Saharan Africa (SSA) and Asia [4]. For the 2.8 billion affected people, procuring biomass fuels and using inefficient cooking stoves also imposes other burdens [4]. This includes for example, a time burden on families, especially women and children [5], who are responsible for most households' cooking and fuel collection. This asymmetric burden contributes to gender inequality. In some locations, namely heavily charcoal-dependent areas in urban SSA, more efficient technologies would also reduce the amount of biomass fuel that must be purchased [6]. Finally, traditional cooking causes severe environmental harms, in the form of unsustainable harvesting of woodfuels that adds to forest degradation and climate change [7], and emissions of black carbon, a powerful climate-forcing agent itself [8]. Hence, cooking with polluting and inefficient technology is associated with negative health, social and environmental outcomes, making it a major challenge for sustainable development.

National policies promoting clean fuels and technologies are essential to achieving Sustainable Development Goal (SDG) 7 on universal access to affordable, reliable, sustainable and modern energy. In HAP-exposed regions, adoption rates of existing clean cooking interventions such as improved cookstoves (ICS) and clean fuels (e.g., liquefied petroleum gas or LPG, ethanol) has historically been slow [4,9]. Indeed, while the global proportion of households dependent on polluting fuels has been decreasing over time, due to high population growth, absolute numbers are flat, and progress is lagging behind that for many other SDG targets, including access to electricity [4]. Among the reasons for the low uptake of cleaner cooking options is their prohibitive upfront or running costs [10], strong ingrained preferences for traditional cooking methods [11,12], and thin markets or unreliable supply chains [13,14]. Even among adopters of transitional (e.g., ICS) and clean technologies, stove stacking (i.e., continued use of traditional cooking methods) remains common [15].

In addition to promoting clean cooking options, studies have argued for design-based (e.g., improving kitchen ventilation, reducing time spent near cooking stove or fire) and policy (i.e., financial and regulatory) interventions to reduce HAP exposure [16,17]. There is growing consensus that a range of policy instruments may be needed to shift behaviors favoring households' movement up the energy stack [9,18]. Some examples of policy interventions to support the clean energy transition are subsidies, innovative business models, limitations or bans on specific polluting fuels (e.g., charcoal), consumer financing and large-scale behavior change campaigns [19].

This paper describes a new tool (dubbed the 'benefits of action to reduce household air pollution' or BAR-HAP Tool) that quantifies the total net health and economic benefits of the various policy actions listed above, as they pertain to specific technology transitions (e.g., from traditional biomass stoves to LPG or improved biomass-burning forced-draft stoves). BAR-HAP is one of the critical elements of the WHO's Clean Household Energy Solutions Toolkit (CHEST, available at: https://www.who.int/airpollution/household/chest/en/), a set of resources that enable countries to develop policies for expanding clean household energy use.

More specifically, BAR-HAP builds on a prior cost-benefit model that was aimed at describing the household net benefits of cooking transitions [20]. That framework was expanded to allow calculation of the public and private costs and benefits of clean cooking options at a national and sub-national level.

This paper's scientific contribution is first, to describe the underlying structure and methods incorporated into the BAR-HAP Tool. Second, it presents an illustrative application of the tool that demonstrates the model's potential usefulness, for practitioners, in providing a holistic appreciation of the implications of inefficient cooking technology, and for assessment of the pros and cons of different intervention strategies to address this energy poverty problem. BAR-HAP is very different from other energy sector planning models, such as the Long-range Energy Alternatives Planning-Integrated Benefits Calculator (LEAP-IBC) and the Combined Heat and Power (CHP) Energy and Emissions Savings Calculator. LEAP-IBC, for instance, helps governments compute the greenhouse gases and short-lived climate pollutants and develop mitigation scenarios [21]. The CHP Energy and Emissions Savings Calculator meanwhile quantifies and "compares the estimated fuel consumption and air pollution emissions ($CO_2e$, $SO_2$ and $NO_X$) of a CHP system and comparable separate heat and power system (e.g. grid power)" [22]. In contrast, the BAR-HAP Tool calculates the private net benefits (time savings, fuel savings, morbidity reductions and mortality reductions) and social net benefits (all private net benefits, along with community spillovers for health benefits, climate and environmental benefits) of potential cooking transitions within a given context, under real-world policy interventions. As became evident during piloting in Nepal, the tool facilitates interactions between relevant stakeholders–researchers, policy-makers outside of the health sector, and implementers of clean energy interventions–because it allows calculation of the government, household, and broader societal costs and benefits of improved cooking policies. Its aim is to support decision-making on ways to accelerate clean energy transitions at national or sub-national scales, by linking technology adoption and outcomes to the effectiveness of specific policy interventions.

## 2. Background: Effectiveness of policy interventions to reduce HAP

This section describes empirical evidence on the effectiveness of five policy instruments included in the BAR-HAP Tool, that are intended to reduce exposure to HAP, namely: stove subsidy, fuel subsidy, stove finance, fuel bans and behavior change communication (BCC). These interventions were selected because they comprise the most commonly-deployed policy instruments that aim to increase household adoption of cleaner technologies, as opposed to other interventions such as certification and standards that cannot be directly linked to technology use. In the clean cooking sector, arguments have been made for subsidies, market development, and awareness creation [23]. However, empirical studies on the impacts of such instruments are extremely limited. Several recent experimental studies have shown ICS demand to be strongly responsive to price subsidies, financing, and supply chain development [14]. Financing by itself also boosts demand, while BCC appears to have a modest influence on willingness to pay [10]. Combining a free trial, time payments and a chance to return improved charcoal stoves (allowing risk-free returns if the technology was found to be undesirable) relaxed rural Ugandan households' liquidity constraints [24]. There is evidence from Cambodia that economic incentives (i.e., stove use subsidies and rebates) facilitate initial ICS adoption but do not necessarily induce long-term use [25].

At a macro-scale, the Indonesian government increased subsidies for LPG stove and fuel and concurrently reduced subsidies for kerosene to encourage the transition to clean cooking

[26], with much success [27]. In Ecuador, LPG subsidies have facilitated transition from traditional fuels to LPG [28], but many rural households continue to use woodfuel alongside LPG and induction stoves [29]. In measuring the costs of policies to provide clean cooking (i.e., LPG) to South Asians, Cameron et al. [30] argue that the most cost-effective policies typically require large stove subsidies.

Prior studies have used descriptive cost-benefit simulations to show that stove subsidies make cleaner cooking options considerably more attractive to users, but that the private net benefits of transitioning may still remain negative for many technologies and households [20]. This implies that even with free distribution of stoves, socially optimal take-up and use may not be achieved. Urban charcoal users, meanwhile, appear highly price sensitive: There is evidence from Ethiopia that charcoal demand had the highest own-price elasticity, and had high cross-price elasticities with firewood [31]. Studies have argued that LPG price subsidies could reduce firewood use [32], but LPG is much more likely to be used by the rich, unless specifically targeted to low-income populations [33]. Owing to cross-price elasticities, fuel subsidies in India would be less likely to reduce demand for polluting fuels like coal and firewood, and improved LPG availability and HAP awareness would be needed to increase demand [34]. Relevant evidence from India's recent LPG subsidy expansion program indicates a large increase in the number of LPG users, but that LPG refill rates have fallen short of the levels needed to achieve the substantial HAP reductions needed to improve health [35].

Heavier regulatory action (e.g., fuel use bans) has also been attempted for some fuels in some contexts. To streamline and sustainably manage non-industrial charcoal production in Kenya (where 40% of the total population uses charcoal stoves for cooking), various restrictions have been attempted, including the Forest (Charcoal) Regulations 2009 and Gazette notice of February 2018 [36–38]. However, neither has been effective and concerns have been raised about negative impacts on the poor and those working in Kenya's large charcoal industry. Under China's coal-to-electricity program that bans coal and provides subsidies for electricity and electric heat pumps, households in high- and middle-income districts have largely eliminated coal use. In low-income districts, however, poor households continue to rely on the polluting fuel [39]. Policies encouraging clean fuel use are often not targeted to low-income households [40].

BCC campaigns are widely deployed to improve public health and development in LMICs, for example for HIV prevention, reproductive health and family planning, reducing malaria, improving sanitation and child survival, as well as for various community empowerment programs [41,42]. There is mixed evidence of impacts of similar BCC techniques (e.g., shaping knowledge, reward and threat, social support, comparisons) on clean stove uptake, health and environmental outcomes, as there is a lack of good quality data and uniformity across study designs also hinder meta-analysis in this domain [43]. As noted earlier, a small (roughly 10%) increase in demand for cleaner technology was found in Uganda following exposure to health-related and other marketing messages [10].

## 3. Methods

This section presents the conceptual model behind the BAR-HAP Tool, followed by a description of the piloting and revision process that was used to improve it, an overview of its advantages and disadvantages, and finally a brief summary of the data sources providing its default parameterization.

### 3.1 Conceptual framework

The BAR-HAP Tool considers (1) a series of technology and fuel transitions from more polluting to cleaner options, as encouraged by (2) a set of policy interventions. Given these choices,

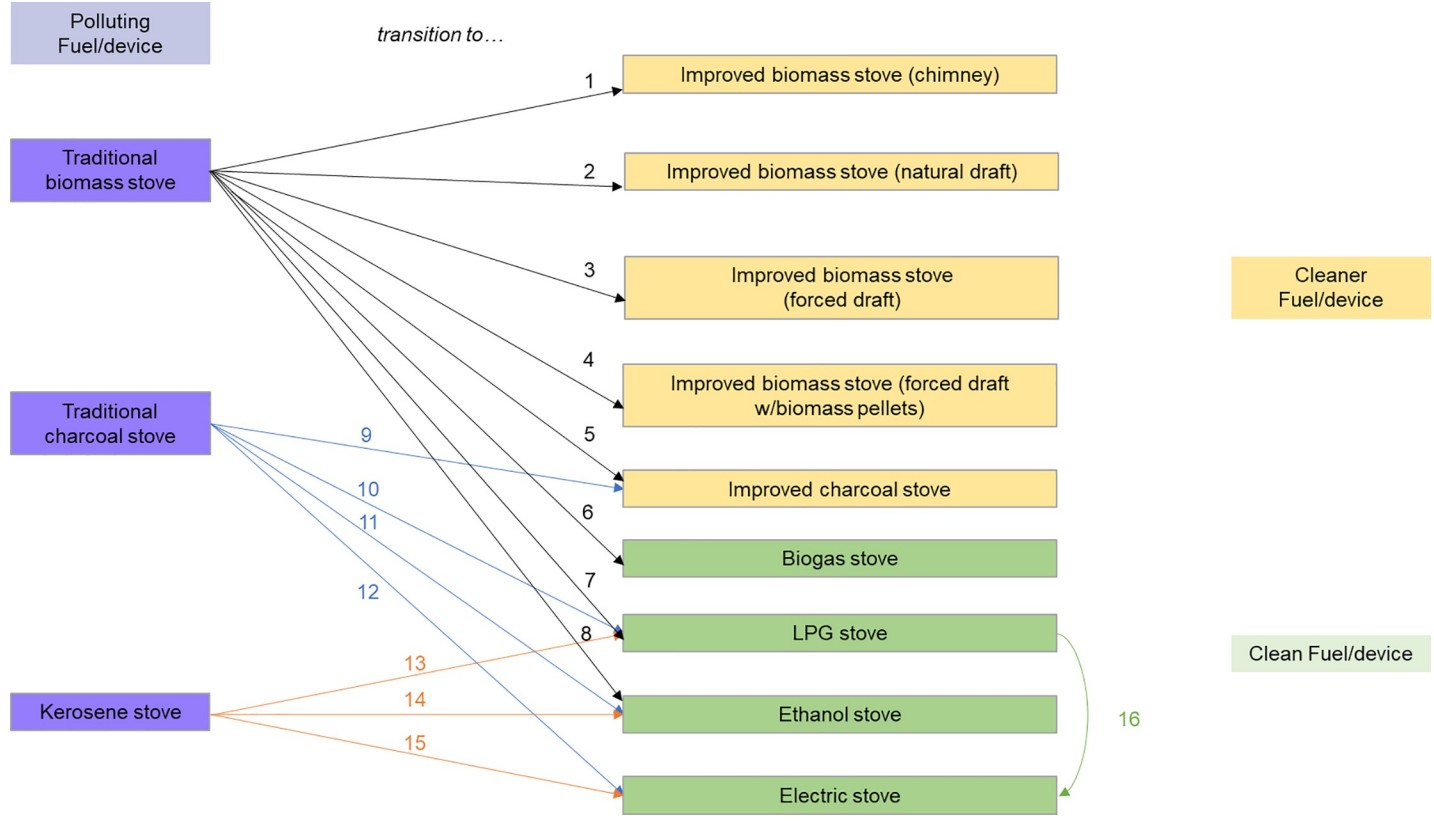

**Fig 1. The sixteen cooking technology or cooking fuel transition scenarios included in the BAR-HAP tool.**

the tool outputs quantified and monetized measures of (1) government costs (which allows estimation of the scale of public financing burden)–overall and on a per capita basis, net of public cost of illness savings; (2) the disease burden averted (disability-adjusted life years or DALYs avoided) and the cost effectiveness ratio (total public cost in US$/total DALYs avoided); and (3) net social benefits, in net present value (NPV) terms. Thus, one can determine whether public financing, health, and full economic objectives are aligned given the selected policy instruments and target transitions. Detailed equations are provided in the WHO BAR-HAP User Manual [44], available at: https://www.who.int/airpollution/household/interventions/chest-module3-BAR-HAP-tool/en/#:~:text=The%20WHO%20Benefits%20of%20Action,cooking%2Drelated%20household%20air%20pollution.

## 3.2 Cooking transitions

The set of 16 fuel and technology transitions included in the tool were selected based on technologies that are currently available in the global market as well as the feasibility of supporting specific technologies. These include clean fuel and technology combinations, which are defined based on the WHO Guidelines [3] and focused on the health benefits of HAP reduction. The transitions also include transitional technologies, which are those that provide substantial benefits but do not reach WHO Guidelines levels. The technology/fuel transitions are classified into four major types (Fig 1), from:

a. Biomass to so-called transitional fuels and technologies;

b. Biomass to clean fuels and technologies;

c. Kerosene to clean fuels and technologies; and

d. One clean fuel/technology to another (specifically LPG to electric). [Note: This transition was included because several countries, including the BAR-HAP Tool pilot country (Nepal), are interested in decreasing their reliance on imported gas, given their ability to generate electricity locally].

### 3.3 Policy instruments

Based on the existing literature and recent policy discussions, we include five policy instruments (or packages of instruments) that could facilitate a clean cooking energy transition (Fig 2):

a. Subsidy for stoves only;

b. Subsidy for fuel (where fuel subsidy is only possible for biomass pellets, LPG, electricity and ethanol), alone or in concert with stove subsidy;

c. Stove financing that would allow adopting households to spread payments for new technology over time, alone or in concert with stove subsidy;

d. BCC, alone or in concert with stove subsidy; and

e. Technology ban.

The consequences of implementing these policies can be considered for each of the sixteen transitions listed above, or for combinations of them, but the BAR-HAP Tool only allows for selection of one of the five policy actions detailed above per cooking transition, to avoid users selecting duplicative interventions. In other words, stove subsidy can be combined with fuel subsidy, financing, or BCC interventions, but the latter three cannot be combined together at this time, largely owing to lack of evidence on the effects that such combined interventions would have. We also note that modeling the social net benefits of a fuel ban requires data on the demand for different fuel types, and derivation of the net consumer surplus that would be lost from forcing fuel switching based on that demand relationship. The BAR-HAP Tool currently incorporates illustrative data on the demand for different fuels that leverages primary data collected in Kenya and Nepal in 2019.

When multiple transitions from a single technology are considered (for example, transitioning some users from traditional biomass stoves to simple natural draft stoves (transition 1) and some users to LPG stoves (transition 5)), the model requests the user to specify the relative percentage of traditional users targeted for each transition. This allowance for "laddering" of transitions provides flexibility for analyses especially in locations where transitional technologies will remain relevant for some time. It also allows for targeting less than the entire population, if some segments are likely to be especially hard to reach (Section 5 provides an example of a laddering case).

### 3.4 Costs and benefits included

Using the WHO Non-Communicable Diseases (NCDs) Costing Tool costing framework (available here: https://www.who.int/ncds/management/c_NCDs_costing_estimation_tool_user_manual.pdf?ua=1), our tool tracks the costs to the (i) government (from personnel/staffing, training, advertising, equipment, subsidies, program costs) and (ii) beneficiaries (stove

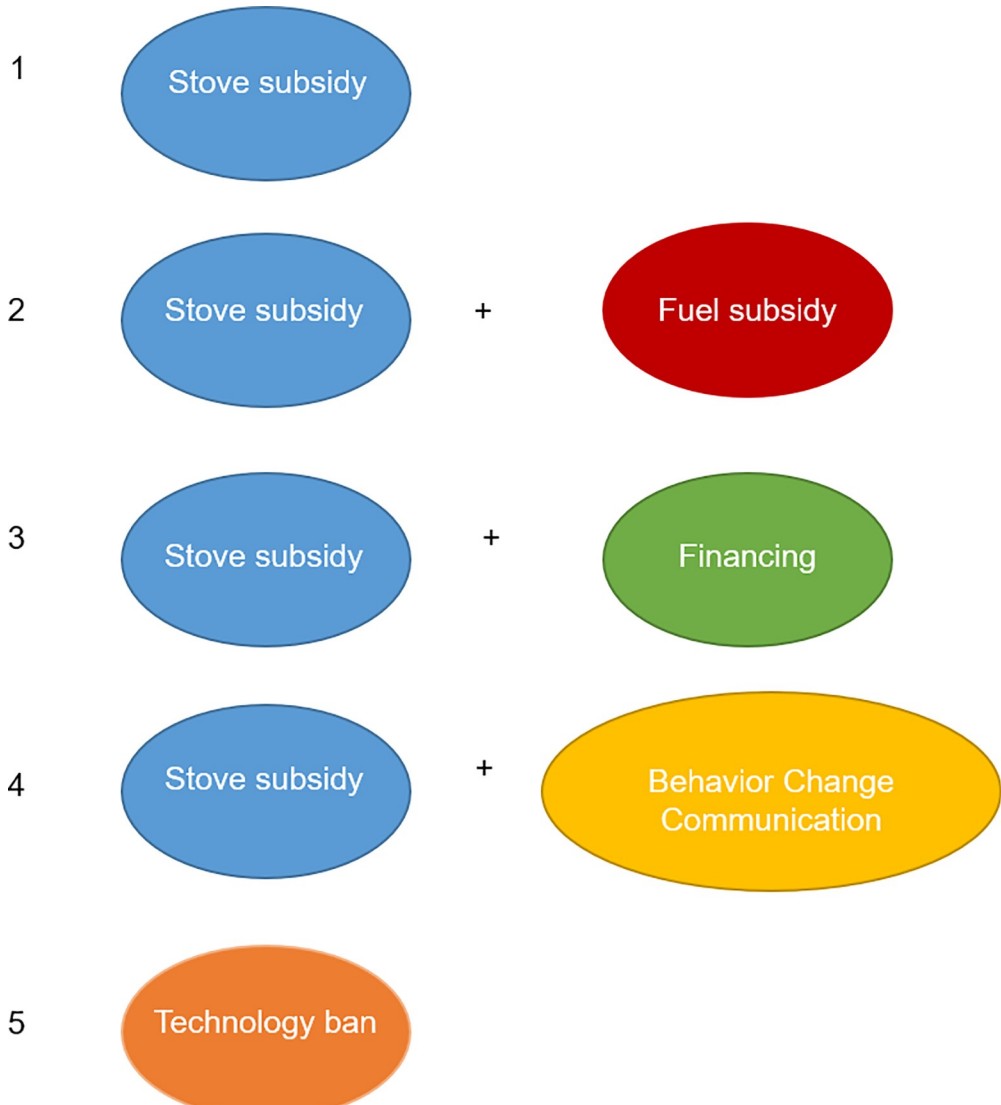

**Fig 2. Five policy interventions that can be applied to all cooking fuel or technology transition scenarios.**

and net fuel costs after subsidies, including valuation of fuel collection time; learning cost; maintenance costs) [45]. On the benefits side (note that this aspect was not included in the WHO NCDs Costing Tool), our tool quantifies and monetizes (i) health improvements (reduced morbidity and mortality)–both to adopting households themselves and to society (including spillovers to improved ambient air quality); (ii) cooking time savings; and (iii) environmental benefits (climate mitigation and reduced ecosystem pressure).

The typology of costs and benefits included in the tool (Table 1) is taken from prior work on calculating the net benefits of cleaner cooking in energy-poor settings [20,46]. In addition to the four health conditions included in Jeuland et al. [20] (ALRI, COPD, IHD and lung cancer), we also include stroke. Though Burnett et al. [47] did not consider stroke in their integrated HAP dose-response function for $PM_{2.5}$, we have assumed that the reductions in stroke would be comparable to the least reduced of the other four included health consequences (which depends on the initial HAP concentration that is specified). This is likely a conservative

**Table 1. Typology of costs and benefits included in the BAR-HAP tool.**

| Costs | |
|---|---|
| 1. Government subsidy costs<br>(i) Stove subsidy cost<br>(ii) Fuel subsidy<br>(iii) Program costs | 2. Private costs<br>(i) Stove cost<br>(ii) Fuel cost, pecuniary and non-pecuniary, e.g., collection time cost<br>(iii) Maintenance cost<br>(iv) Learning costs |
| **Benefits** | |
| 1. Private health benefits<br>(i) Morbidity reductions of chronic obstructive pulmonary disease (COPD)<br>(ii) Mortality reductions of COPD<br>(iii) Morbidity reductions of acute lower respiratory infections (ALRI)<br>(iv) Mortality reductions of ALRI<br>(v) Morbidity reductions of ischemic heart disease (IHD)<br>(vi) Mortality reductions of IHD<br>(vii) Morbidity reductions of lung cancer (LC)<br>(viii) Mortality reductions of LC<br>(ix) Morbidity reductions of stroke<br>(x)Mortality reductions of stroke | 2. Social health benefits (incorporating HAP contribution to ambient air pollution)<br>(i)Morbidity reductions of COPD, ALRI, IHD, LC and stroke–using social discount rate and accounting for health spillovers<br>(ii)Mortality reductions of COPD, ALRI, IHD, LC and stroke–using social discount rate and accounting for health spillovers |
| 3. Time savings | 4. Basic (Kyoto-protocol gases) and full (with additional pollutants) climate benefits |
| 5. Other environmental benefits (sustainability of biomass harvesting) | |

assumption, though we highlight the lack of formal modeling evidence supporting specific reductions in stroke from mitigating HAP. It is also worth highlighting that other health conditions that are potentially related to traditional cooking are not included due to lack of sufficiently strong and consistent evidence.

## 3.5 Model assumptions

As in any cost-benefit analysis, the BAR-HAP Tool rests on a number of assumptions, most of which can be modified by users in accordance with the data available in their locations. First, the model comes with a default set of 331 user-modifiable parameters that figure into the calculations (default assumptions for these parameters and data sources informing them are included in the user manual referred to in footnote 4). This large number of unique parameters is partly due to their differentiation by transition, as many pertain to the same concepts (for example the technology-specific price at which uptake of a stove begins). In its current version, the tool is parameterized primarily with inputs from Nepal- or South Asia-specific studies, though global estimates are used where regional data are unavailable. Similar to the WHO's NCDs Costing Tool, BAR-HAP is also pre-filled with country-specific default demographic data and epidemiological data from global sources (see data description below). The full set of input parameters can be categorized as follows:

1. Demographic parameters: Total population, average household size and average number of children under five per household.

2. General economic parameters: These include factors like the social discount rate, and valuation parameters such as the shadow value of time spent cooking (as a fraction), the unskilled wage rate, and the value of morbidity and mortality reductions.

3. Baseline cooking parameters: Examples of these include the proportion of the population using various traditional, transitional, or clean fuels, the time spent cooking on traditional stoves, the fuel spent cooking on traditional stoves, and the fuel collection time.

4. Demand and intervention parameters: Strategies and interventions to support each selected transition; stove demand parameters; subsidy amounts and leakages; implementation costs and effectiveness parameters.

5. Stove and fuel parameters: Examples of these are the costs of stoves and fuels, the efficiencies and emissions of stoves, learning hours and maintenance costs.

6. Health parameters: The prevalence or incidence rates and mortality rates of the five included health outcomes, expected life expectancy remaining for those specific conditions, DALY weights, exposure adjustment factors, extent of spillovers to ambient air pollution, lagged health impacts, and parameters for calculating relative risks and population attributable fractions of the HAP-related diseases (specifically, we account for the effect of $PM_{2.5}$ on the following five health outcomes: COPD, ALRI, IHD, lung cancer and stroke).

7. Environmental parameters: Tree replacement cost, social cost of carbon, parameters related to the global warming potential of relevant pollutants (which include $CO_2$, CO, $N_2O$, $CH_4$, BC and OC, where the latter is a net cooling agent). [Note that black carbon is included in $PM_{2.5}$, due to its very small size, but, consistent with the environmental health literature, we do not differentiate the health impacts of black carbon versus the other constituents of $PM_{2.5}$].

Second, as discussed above, it is not possible with the current set up to include more than one of the financing, fuel subsidy, and BCC interventions within a specific transition, though stove subsidy is possible to include alongside each of these other policy interventions. This decision was made due to a lack of evidence on how the effects of multiple interventions would compound to affect both costs and benefits. Specifically, there is little evidence to suggest whether impacts are additive or multiplicative, or whether costs scale proportionately to the way they do for individual interventions.

Third, a key set of assumptions constrains the demand for improved fuel/technology and the way that interventions then affect that demand. Here, the default parameterization of the model is based on the very limited high-quality evidence reviewed above in Section 2. The demand for improved stoves is assumed to be linear, based on results from Pattanayak et al. [14] which showed a highly linear and price elastic response of demand. Three parameters can be specified: (a) the maximum price that anyone would pay for a stove; (b) the maximum coverage rate that can be achieved (i.e., the maximum percentage of households that would use the selected technology/fuel); and (c) the price at which coverage would reach that maximum. Default levels for these are based on the evidence in Pattanayak et al. [14] and Nepal-specific data for biogas technology. Evidence for a fuel subsidy among relevant populations is even more scant; the model thus sets a price elasticity of -1 such that a 50% reduction in the cost to users effectively doubles the uptake of the technology. This assumption, like that related to the shape of the demand curve for cleaner stoves, can also be modified if desired. Based on these demand curves, user-specified stove and fuel subsidy adjustments lead to changes in uptake and therefore generation of benefits.

Fourth, the effects of financing and BCC interventions are assumed to boost the total willingness to pay for technology by 40% and 10%, respectively, based on the evidence, reviewed above [10]. These authors found that "time payments" increased total WTP (or the area under the demand curve) by 40%, and that health messaging increased demand by about 10%. Both of these parameters–which are adjustable in the model–thus cause a pivoting outward of the

linear stove demand curve that allows adoption to begin at a higher initial price, and that increases the area under the demand curve proportionately to these increases. Fifth, we include a variable for usage rate of improved technologies indicating that use is expected to be less than complete, acknowledging that stove stacking is the norm [11,15]. This usage rate is critical in moderating the benefits from interventions to reduce HAP [46], as it affects household emissions and net fuel and time savings, and therefore also health and environmental benefits. Sixth, to account for the increased exposure from behavioral adjustments that accompany the change in cooking fuels and technologies [48], and specifically the fact that cooks exert less effort to avoid HAP as the cooking environment becomes cleaner, we include an exposure adjustment parameter. This factor accounts for the difference between kitchen concentrations and personal exposures that may arise from behavioral and other factors that lead to the divergence of the two measures. Seventh, in the calculation of cost effectiveness metrics, age weighting was not used and results are not disaggregated by age groups. As such, life years lost from mortality were based on the average life expectancy for those affected by the specific health conditions included in the model, and these were discounted using the same social discount rate as that used to calculate overall NPV and weight the social cost of mitigation of different climate forcing agents.

Finally, the model aggregates costs and benefits (and other outcome metrics) over a 15-year time horizon. In the default set-up, the initial five years are treated as a ramp up period, during which the intervention planning proceeds for two years without generating benefits, followed by three years over which a third of the target population is added in each year. Starting in year 6, then, the full target population is assumed to be affected. Alternatively, the policy interventions can be considered fully implemented starting in the first year, in which case benefits begin immediately.

## 3.6 Model piloting and testing

To demonstrate the functionality and types of results that can be produced using the BAR--HAP model, we developed and piloted an application to Nepal. Though Nepal presents an important location to test policy interventions to address the challenge of clean cooking transition, this application is not intended to provide a complete policy analysis for that setting. The setting is relevant because Nepal ranks among the lowest three countries on the Environmental Performance Index's global air quality rankings, as measured by the indicators on household solid fuels, $PM_{2.5}$ exposure and ozone exposure [49]; there is predominant reliance on solid fuels for cooking, such as firewood and cow dung [50]; (c) HAP-attributable deaths are high (21,603 as of 2019) [51]; and the country is facing rapid deforestation [52,53].

To support the piloting, a 3-day pilot workshop was held outside of Kathmandu with a variety of stakeholders from the national government, local researchers, professionals working in the health system, and development partners working on health and energy issues. We did not collect primary data, but rather relied on secondary data to parameterize the model for this setting, as described in the section '3.8. Data sources'. Based on user feedback, two new transitions were included in the model (from traditional biomass to biogas cooking, and from LPG to electric cooking), as well as additional functionality to allow selection of multiple transitions from a single technology (the "laddering" aspect of the tool). Other feedback helped to inform the finalization of model summary graphs and figures, and parameterization to better reflect the realities of the context in Nepal.

## 3.7 Advantages and disadvantages of the BAR-HAP Tool

This decision support tool has three principal advantages:

1. It provides a user-friendly, convenient framework for estimating a set of costs and benefits that a) is more extensive, and b) applies to a wider range of single or multiple-technology cooking transitions (implemented at sub-national or national scales) than can be found in any other comparable planning tool.

2. Users can modify an extensive range of input parameters based on their knowledge of the best available context-specific data, to develop predictions of the impacts of policy instruments tailored to their location and baseline situations, acknowledging that households do not always make a complete switch to cleaner options.

3. It can help inform decision-making by providing a range of outputs that may be weighted according to national or regional priorities for maximizing health or development impacts, working within budget constraints.

The BAR-HAP Tool also has important limitations, some of which could be addressed in future work to improve the model:

1. In its current form, the model is static. It thus does not account for dynamics that increase or decrease population, rates of technology adoption and changes in affordability, or the prevalence or incidence rates of diseases over time.

2. For the most part, the tool does not factor in changes in health sector implementation costs as the scale of intervention provision (either economies or diseconomies of scale) increases. Only the stove and fuel costs, and promotion program costs scale based on the number of users targeted.

3. While we include an exposure adjustment factor, we do not incorporate structural characteristics of the household (for which data may not exist at the country level), thereby not considering cross-household heterogeneity.

4. The contribution of ambient air pollution, which could nullify the effects of HAP reductions from clean cooking, is not accounted for.

5. Consumers' preferences for improved and clean cooking scenarios and related policy interventions are not incorporated. A welfare-theoretic perspective on private benefits, for example, would equate these to the area under the demand curve, but BAR-HAP calculations of these private benefits are rather based on valuation equations that pertain to the specific benefits presented in Table 1. These may diverge from individuals' willingness to pay for those improvements for a range of reasons.

6. Though their addition would not be a simple task manageable by the majority of users, additional transitions between cooking fuels and technologies could be incorporated into the tool, which is currently limited to the sixteen described above, and as the piloting in Nepal revealed, users in specific contexts are likely to have particular interest in certain types of transitions that may not be included in the current version.

## 3.8 Data sources

Public data sources provided the data used for most of the model parameters, though some data specific to Nepal were obtained from stakeholders. Government personnel, legal, training, media/advocacy and materials costs needed for supervision and program implementation, were sourced directly from the prior WHO NCDs Costing Tool [54–56], which implements a unit costs approach. Country-specific epidemiological data for the five conditions were sourced from the Global Burden of Disease project [51], as were DALY weights for morbidity.

Most other parameters–baseline cooking details, stove and fuel costs and characteristics, behavioral parameters, economic valuation parameters–were taken directly from the cost-benefit model and database in Jeuland et al. [20]. PM emissions and exposure adjustment factors were updated based on a recent review [57]. Cost-of-illness for stroke, which was not included in the latter, was taken from a recent systematic review of LMIC evidence [58]. Where possible, these parameters were adjusted for relevance to the Nepal case, using news sources (Kathmandu Post), Nepal Census 2011 data, and local stakeholder data, for example for the cost of chimney and biogas stove technologies. All parameters can be found in the BAR-HAP model, and sources are described in the appendix of the user manual for the tool.

## 4. Results: Example applications for the case of Nepal

This section presents illustrative results of two distinct transitions–selected to demonstrate the functionality and insights that can emerge from use of BAR-HAP–that are relevant for the context of Nepal. Though many of the input parameters have been validated for Nepal, some have not, such that the results should not be taken as justification for specific policy-making in the country. The two transitions considered are the following:

1. A transition of all users from using firewood with traditional stoves to LPG stoves; and

2. A hybrid transition of half of the users from using firewood with traditional stoves to improved natural draft biomass stoves, and of the other half of these users from traditional stoves to LPG stoves

These two potential transitions affect the greatest number of people in Nepal, since 74% of households are estimated to use traditional firewood-burning stoves [50]. In what follows, we first present results for various levels of stove subsidies, and then compare these with the outcomes from other interventions.

### 4. 1 Transition 1: Traditional biomass stoves to LPG stoves

We begin analysis of this transition under a policy of stove subsidy alone, and vary that subsidy from 0% to 100% to demonstrate how this affects the following outcomes: a) the present value (PV) of the net public costs of the intervention; b) the per capita public cost of the intervention; c) the present value of public cost-of-illness savings; d) the total disease burden avoided (DALYs avoided); e) the cost effectiveness ratio (in US$/DALY avoided); and f) the present value of the total social net benefits of the intervention. Results are shown in Table 2.

Table 2. Outcomes for transition from traditional biomass stoves to LPG stoves in Nepal, as a function of varying stove subsidy.

| Outcome | Stove subsidy level | | | | |
|---|---|---|---|---|---|
| | **0%** | **60%** | **70%** | **85%** | **100%** |
| Target population covered | 0% | 0% | 31.7% | 81.2% | 100% |
| PV Net Public Cost (millions of US$) | 7.8 | 7.8 | 86.1 | 246.3 | 348.1 |
| Per capita public cost | 0.36 | 0.36 | 4.05 | 11.61 | 16.44 |
| PV Public COI savings (millions of US$) | 0 | 0 | 25.6 | 65.4 | 80.6 |
| DALYs avoided | 0 | 0 | 172,943 | 442,735 | 545,118 |
| CER (PV$/DALY avoided) | n.a. | n.a. | 674 | 756 | 869 |
| PV Social Net Benefits (millions of US$) | -7.8 | -7.8 | 381.6 | 981.3 | 1,200.7 |

Notes: All assumptions other than the subsidy level are left at their default BAR-HAP values. Abbreviations: PV = Present value, US$ = United States dollars, LPG = Liquified petroleum gas, COI = cost of illness, DALY = disability-adjusted life year, CER = cost effectiveness ratio.

This analysis highlights a few important aspects of this model. First, we observe that the 0% stove subsidy level generates no benefits, and only very minor costs. This is because the demand curve is specified in the model such that no household would adopt the LPG stove at its full default cost of US$39 (this default demand curve, for the simple subsidy as well as the alternative financing and BCC campaign interventions is shown in Fig 3). Nonetheless, the program still incurs fixed costs for administrative personnel, trainings, and other aspects paid by the government. The same applies even at a 60% subsidy, which remains insufficient to spur adoption. Starting with a 70% subsidy and above, however, the price drops below the level needed for some households to adopt the technology. In addition, the costs to the government increase sharply as adoption and subsidies increase, driven by both more target beneficiaries taking up the stoves and by the increasing share of the cost that is paid by the government, despite significant public cost of illness (COI) savings. Free provision is estimated to cost less than half a billion US$ over the 15-year promotion period (equivalent to a one-time cost of 1.2% of annual GDP in Nepal), assuming that stoves are replaced at the end of their useful lifespan. At the same time, benefits also increase with adoption, and exceed costs in present value terms at all subsidy levels above 61%, which is the breakeven subsidy level from a social perspective. The relative proportions of different costs and benefits, which remain similar across stove subsidy levels (but for which we show the 70% subsidy case), indicates that climate mitigation provides the largest category of benefits, followed by avoided mortality, household time savings, and finally other ecosystem benefits and avoided morbidity (Fig 4). The cost effectiveness ratios (CER) range between about 674 and 869 US$/DALY avoided across subsidy levels. The largest cost items are stove costs (spread over the government and users), program implementation costs (for stove distribution), and technology maintenance costs.

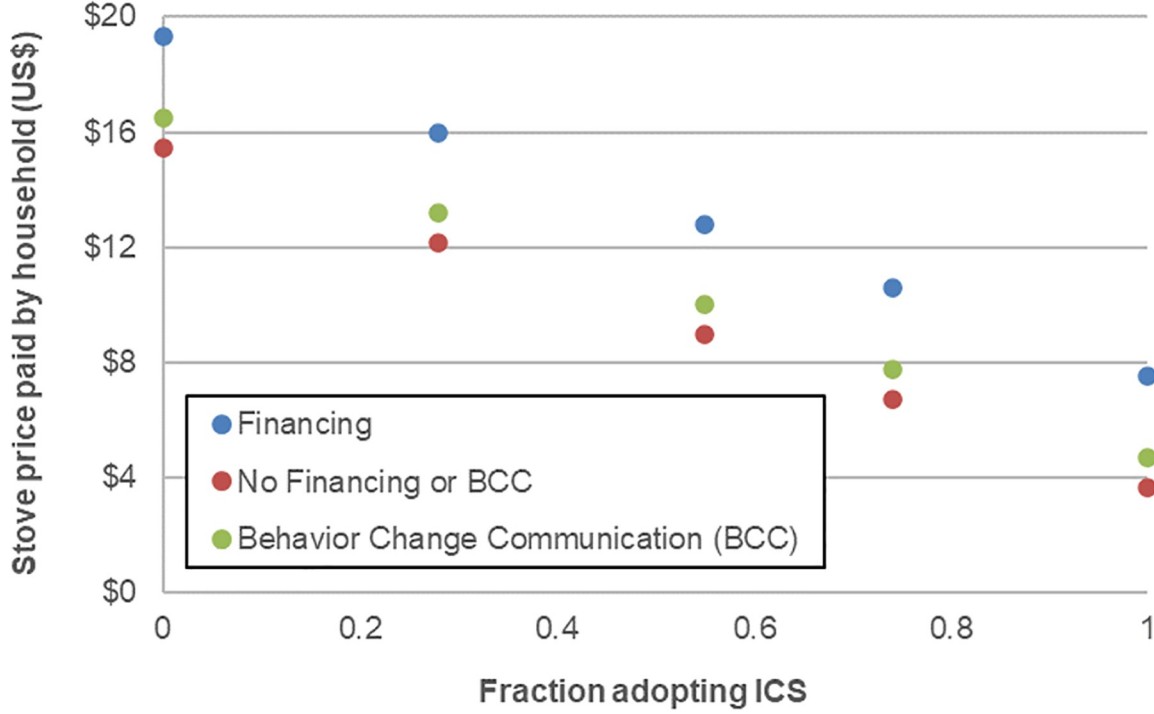

**Fig 3. Default assumptions for demand for improved cookstoves.**

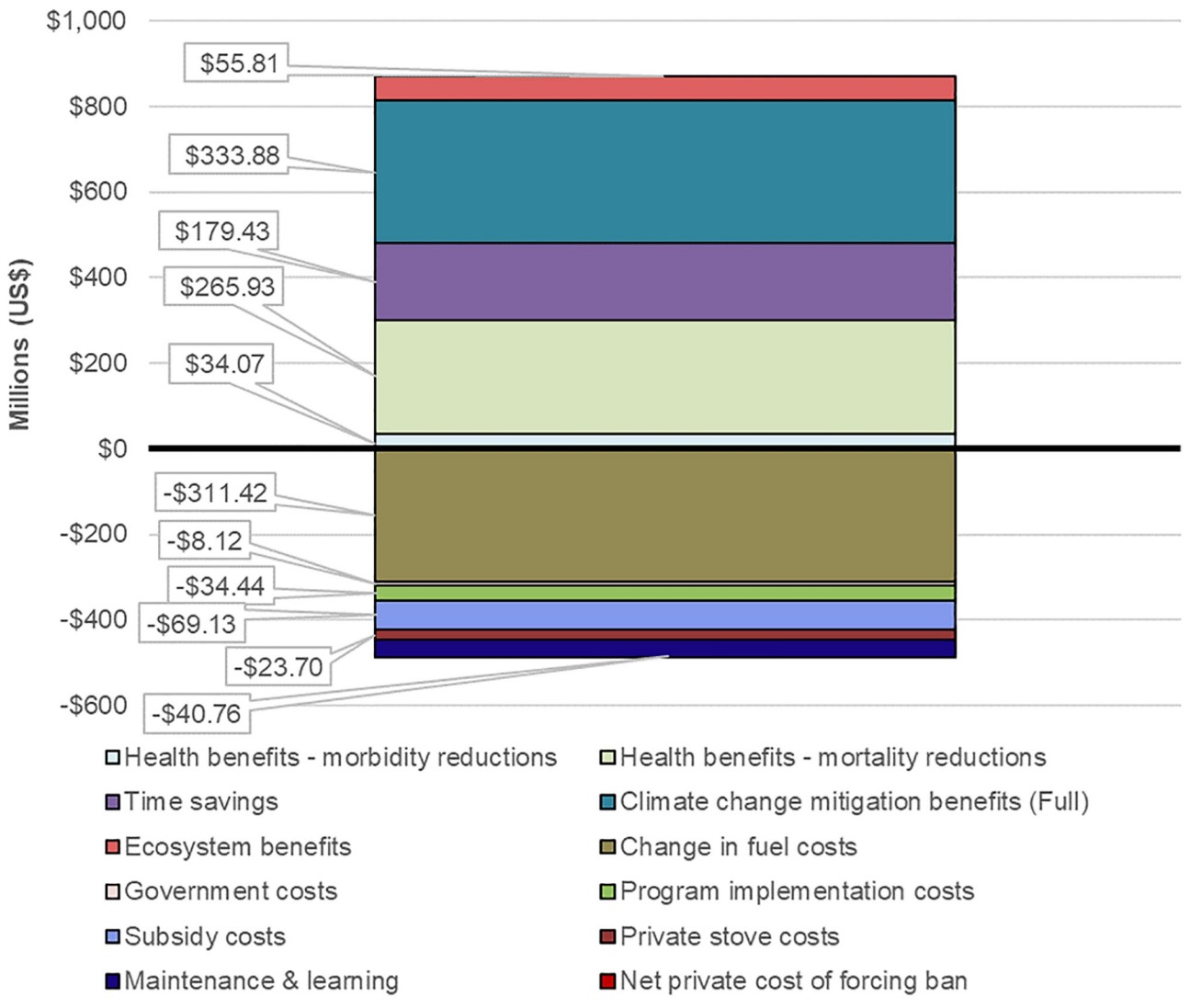

**Fig 4. Breakdown of total present value of costs and benefits (in US$), with 70% stove subsidy for shift from traditional biomass stoves to LPG stoves.**

Table 3 summarizes these outcomes for the other three relevant interventions (not including a ban on firewood use, which is impractical in Nepal): addition of financing for stoves, fuel subsidy for LPG, and BCC. Stove financing boosts demand by 40%; this leads to considerably faster adoption and also implies somewhat higher costs (in fact the whole target population adopts the technology at an 85% subsidy under this policy). Nonetheless, financing appears more cost effective and more cost beneficial than the stove subsidy-only policy because additional LPG adoption generates net social benefits (again mainly cooking and fuel collection time savings, and climate mitigation; the latter benefit category accrues to society as a whole). A 25% fuel subsidy for LPG is shown to be much more costly than the other policies, peaking at a one-time equivalent of 7% of annual GDP over the 15-year time horizon when coupled with free stove distribution. As shown in Table 3, this is driven by increasing costs even when no new households switch to LPG (since these fuel subsidies then flow entirely to existing LPG stove owners), and from greater expenses among households once new adoption ticks up due to subsidization of LPG stoves. Due to these higher costs, the fuel subsidy policy is

**Table 3. Outcomes for transition from traditional biomass stoves to LPG stoves in Nepal, as a function of varying stove subsidy.**

| Outcome | Stove subsidy level | | | | |
|---|---|---|---|---|---|
| | **0%** | **60%** | **70%** | **85%** | **100%** |
| With 25% fuel subsidy | | | | | |
| Target population covered | 0% | 0% | 35.7% | 91.4% | 100% |
| PV Net Public Cost (millions of US$) | 512.0 | 512.0 | 937.6 | 1,644.2 | 1,797.8 |
| Per capita public cost | 24.02 | 24.02 | 44.82 | 79.32 | 86.77 |
| PV Public COI savings (millions of US$) | 0 | 0 | 28.7 | 73.6 | 80.6 |
| DALYs avoided | 0 | 0 | 194,561 | 498,077 | 545,117 |
| CER (PV$/DALY avoided) | n.a. | n.a. | 6,637 | 4,588 | 4,586 |
| PV Social Net Benefits (millions of US$) | -512.0 | -512.0 | -83.5 | 576.6 | 586.3 |
| With stove financing | | | | | |
| Target population covered | 0% | 31.0% | 64.0% | 100% | 100% |
| PV Net Public Cost (millions of US$) | 7.8 | 74.7 | 165.9 | 301.4 | 348.1 |
| Per capita public cost | 0.36 | 3.50 | 7.80 | 14.22 | 16.44 |
| PV Public COI savings (millions of US$) | 0 | 25.0 | 51.6 | 80.6 | 80.6 |
| DALYs avoided | 0 | 169,069 | 348,930 | 545,117 | 545,117 |
| CER (PV$/DALY avoided) | n.a. | 597 | 644 | 751 | 869 |
| PV Social Net Benefits (millions of US$) | -7.8 | 371.7 | 772.9 | 1,206.3 | 1,200.7 |
| With BCC | | | | | |
| Target population covered | 0% | 0% | 39.8% | 89.3% | 100% |
| PV Net Public Cost (millions of US$) | 7.8 | 7.8 | 131.5 | 327.0 | 412.0 |
| Per capita public cost | 0.36 | 0.36 | 6.20 | 15.45 | 19.49 |
| PV Public COI savings (millions of US$) | 0 | 0 | 32.1 | 71.9 | 80.6 |
| DALYs avoided | 0 | 0 | 216,940 | 486,732 | 545,117 |
| CER (PV$/DALY avoided) | n.a. | n.a. | 823 | 915 | 1,030 |
| PV Social Net Benefits (millions of US$) | -7.8 | -7.8 | 455.2 | 1,022.6 | 1,136.9 |

Notes: All assumptions other than the subsidy level are left at their default BAR-HAP values. There is no financing cost for the 100% subsidy case. Abbreviations: PV = Present value, US$ = United States dollars, LPG = Liquified petroleum gas, COI = cost of illness, DALY = disability-adjusted life year, CER = cost effectiveness ratio.

considerably less cost effective than the others, though social benefits overall remain positive. Finally, the BCC increases demand and costs much less than the other two policies, and is less cost effective than either the stove subsidy only or stove subsidy plus financing policy for this technology transition, under the default model assumptions.

## 4.2 Transition 2: Traditional biomass stoves to simple improved natural draft stoves (50%) and to LPG stoves (other 50%)

A hybrid transition may be more realistic than the previous transition due to the difficulty of establishing supply chains for LPG in remote rural areas. Transition 2 considers such a hybrid (results appear in Table 4). We ask whether significant economic benefits would be lost relative to the cleaner technology transition option. As shown, cost effectiveness ratios increase; this is the result of reduced DALYs avoided and health benefits. Despite this, total social net benefits are only slightly reduced relative to the LPG only transition, because the natural draft biomass stove still provides significant fuel and time savings, and climate mitigation, previously shown to be important in Fig 3. Benefits can also be generated at somewhat lower subsidy levels, owing to the lower cost of the natural draft stoves. It is important to note that while some

**Table 4. Outcomes for a hybrid transition from traditional biomass stoves to natural draft improved cookstoves (50% of solid fuel users) and LPG stoves (50% of solid fuel users) in Nepal, as a function of varying stove subsidy.**

| Outcome | Stove subsidy level | | | | |
|---|---|---|---|---|---|
| | **0%** | **60%** | **70%** | **85%** | **100%** |
| Stove subsidy only | | | | | |
| Target population covered | 0% | 24.7% | 50.8% | 90.6% | 100% |
| PV Net Public Cost (millions of US$) | 15.5 | 101.5 | 187.0 | 347.8 | 422.1 |
| Per capita public cost | 0.72 | 4.81 | 8.87 | 16.50 | 20.03 |
| PV Public COI savings (millions of US$) | 0 | 4.6 | 19.2 | 42.0 | 49.5 |
| DALYs avoided | 0 | 30,976 | 130,157 | 283,956 | 335,147 |
| CER (PV$/DALY avoided) | 0 | 4,477 | 1,962 | 1,674 | 1,722 |
| PV Social Net Benefits (millions of US$) | -15.5 | 256.0 | 559.7 | 1,019.4 | 1,124.4 |
| With 25% fuel subsidy | | | | | |
| Target population covered | 0% | 24.7% | 52.8% | 95.7% | 100% |
| PV Net Public Cost (millions of US$) | 519.8 | 605.8 | 864.9 | 1,299.0 | 1,399.1 |
| Per capita public cost | 24.38 | 28.47 | 41.08 | 62.18 | 67.02 |
| PV Public COI savings (millions of US$) | 0 | 4.6 | 20.8 | 46.1 | 49.5 |
| DALYs avoided | 0 | 30,977 | 140,965 | 311,627 | 335,147 |
| CER (PV$/DALY avoided) | n.a. | 26,481 | 8,396 | 5,749 | 5,761 |
| PV Social Net Benefits (millions of US$) | -519.8 | -248.3 | 75.1 | 564.9 | 565.0 |
| With stove financing | | | | | |
| Target population covered | 0% | 56.4% | 82.0% | 100% | 100% |
| PV Net Public Cost (millions of US$) | 15.5 | 191.3 | 284.2 | 375.5 | 422.1 |
| Per capita public cost | 0.72 | 9.07 | 13.48 | 17.80 | 20.03 |
| PV Public COI savings (millions of US$) | 0 | 20.1 | 35.0 | 49.5 | 49.5 |
| DALYs avoided | 0 | 135,717 | 237,054 | 335,147 | 335,147 |
| CER (PV$/DALY avoided) | 0 | 1,925 | 1,638 | 1,530 | 1,722 |
| PV Social Net Benefits (millions of US$) | -15.5 | 618.6 | 916.3 | 1,130 | 1,124.4 |
| With BCC | | | | | |
| Target population covered | 0% | 28.8% | 58.9% | 94.7% | 100% |
| PV Net Public Cost (millions of US$) | 15.5 | 145.3 | 265.3 | 440.0 | 505.8 |
| Per capita public cost | 0.72 | 6.90 | 12.60 | 20.89 | 24.03 |
| PV Public COI savings (millions of US$) | 0 | 5.3 | 23.2 | 45.2 | 49.5 |
| DALYs avoided | 0 | 36,028 | 157,207 | 305,954 | 335,147 |
| CER (PV$/DALY avoided) | 0 | 5,521 | 2,308 | 1,967 | 2,065 |
| PV Social Net Benefits (millions of US$) | -15.5 | 270.5 | 600.2 | 988.3 | 1,040.7 |

Notes: All assumptions other than the subsidy level are left at their default BAR-HAP values. There is no financing cost for the 100% subsidy case. Abbreviations: PV = Present value, US$ = United States dollars, LPG = Liquified petroleum gas, COI = cost of illness, DALY = disability-adjusted life year, CER = cost effectiveness ratio.

biomass ICS have higher BC emissions than traditional stoves, such stoves on average still produce lower climate-forcing emissions, for several reasons. First, they reduce biomass consumption, such that any unsustainable harvesting of firewood (which is the norm in most LMICs, see Bailis et al. 2015) leads to lower net $CO_2$ emissions. Second, because fuel consumption decreases, even though the ratio of BC to other PM tends to increase, the overall amount of BC may also decline. This is not always the case, and there is considerable variation across biomass ICS, but our calculation relies on averages across a number of different models, which on average are helpful in reducing overall BC emissions and climate forcing. If users were interested

in particular ICS technologies, the net reductions might not hold, depending on these fuel efficiency and BC emissions characteristics.

## 5. Discussion

This paper's main contribution is in presenting a new decision-support model–the BAR-HAP Tool–that is aimed at helping health and government decision-makers to plan policy interventions to accelerate transitions towards cleaner cooking technologies, and in describing the tool's methodology. The conceptual model behind the tool is based in a framework of costs and benefits, that accounts for partial adoption and use of technology, and incorporates evidence from recent related modeling efforts and field studies. It allows analysis of pricing, financing, and behavior-change interventions, based on published evidence on the effectiveness of these policy instruments. Practical aspects of the tool were modified based on feedback from a diverse stakeholder workshop held with representatives from the national government, local researchers, professionals working in the health system, and development partners working on health and energy issues in Nepal. We also demonstrated here how the tool can be used to consider several cooking transitions in the context of Nepal, to provide a perspective on the types of results that BAR-HAP can generate.

BAR-HAP was designed to be user friendly and is available in the public domain, along with a user manual explaining more of its technical details. It does require significant data input from users to improve its realism and relevance, though national level data are provided in the tool's database whenever possible. Without additional updating of these underlying data, conclusions should be interpreted cautiously, and sensitivity analysis to key uncertainties should be carried out. Complementary efforts to catalog and make consistent data available publicly would considerably strengthen the evidence base underlying the parameterization of the tool. Efforts to provide global datasets with relevant information include the WHO's Global Health Observatory, which contains information on the percentage of households primarily cooking with clean and polluting fuels or technologies and is used for reporting on SDG7; and WHO's Household energy database, which contains nationally-representative survey estimates of the different fuels and technologies households primarily use for cooking, space heating, and lighting [59,60]. Furthermore, users should understand the limitations of the tool, discussed in this article, including the fact that it only covers the contribution of cooking practices to HAP (and does not cover lighting and heating, for example). For the illustrative application to Nepal, preliminary analyses suggest that both "transitional" changes and movements to clean technology are capable of generating significant benefits including health improvements, even if clean technologies generate considerably more of the latter benefits. Still, cost effectiveness ratios–ranging from about US$400 to several thousand US$ per DALY avoided–indicate that transitions to clean cooking are a valuable intervention to improve population health through interventions extending well beyond the health sector. To optimize health benefits, clean cooking should, however, be promoted alongside other public health interventions. Due to its multisectoral nature, the transition to clean household energy does not only produce health benefits but shows also significant co-benefits for climate change mitigation and the promotion of gender and social equality, which is consistent with prior work [20].

The BAR-HAP Tool is part of the WHO's suite of CHEST tools that aim to reduce the burden of disease associated with HAP in LMICs, designed for use by national planners and policymakers. WHO has also developed a HAP module within the OneHealth Tool that is already used for comprehensive health intervention planning [61]. These various efforts will enable the health sector to compare different health, economic and other impacts of clean household

energy transitions, which is critically important given the key role of air pollution as a risk factor for a range of NCDs.

WHO is now working with researchers to expand the database underlying the BAR-HAP Tool to facilitate national and global analyses of policy interventions to spur clean cooking transitions. As part of this expansion, additional training workshops will be organized to catalyze informed decision-making processes, and to support the formulation of new policies appropriate for the countries most affected by the negative health impacts of HAP. In this respect, the calculation of context-specific costs and benefits at household and system levels using national and local data is a key strength that will facilitate tailoring of solutions according to the needs and priorities of particular countries.

In conclusion, tools such as the BAR-HAP Tool can be used to help support decision-making and better understand the myriad ill consequences of traditional cooking. In demonstrating that the energy poverty challenge is multi-faceted and touches health, development, and environmental outcomes, this tool will help facilitate cross-sector dialogue and problem-solving to address this major sustainable development challenge. A longer-term goal of the collaboration that produced this manuscript is to create a global database that would facilitate calculations across LMICs and global regions. The expanded version of the BAR-HAP Tool will take into account the need for additional transitions, and allow calculations by geographic region and comparison across countries and regions, for use by the global health and energy communities. Though beyond the scope of the current article, such comparisons will facilitate identification of the technologies and fuels that are most appropriate given local constraints and realities, as well as targeting of interventions to the countries where potential benefits are most significant.

## Acknowledgments

We are especially grateful to the WHO headquarters and Country Office in Nepal which organized the stakeholder piloting and feedback workshop in partnership with the Ministry of Health and Population in Nepal that helped guide refinement of the BAR-HAP Tool. Those stakeholders provided very valuable feedback, and we are thankful especially to Karuna Bajracharya of the Nepal office for the Clean Cooking Alliance and Samir Thapa at the Alternative Energy Promotion Centre in Nepal, for providing valuable Nepal-specific data for incorporation into the tool.

## Author Contributions

**Conceptualization:** Ipsita Das, Jessica J. Lewis, Heather Adair-Rohani, Marc Jeuland.

**Data curation:** Ipsita Das, Melanie Bertram.

**Formal analysis:** Ipsita Das, Melanie Bertram, Marc Jeuland.

**Funding acquisition:** Marc Jeuland.

**Investigation:** Ipsita Das, Marc Jeuland.

**Methodology:** Ipsita Das, Jessica J. Lewis, Ramona Ludolph, Melanie Bertram, Marc Jeuland.

**Project administration:** Ipsita Das, Ramona Ludolph, Marc Jeuland.

**Resources:** Melanie Bertram, Marc Jeuland.

**Software:** Ipsita Das, Marc Jeuland.

**Supervision:** Jessica J. Lewis, Heather Adair-Rohani, Marc Jeuland.

**Validation:** Jessica J. Lewis, Marc Jeuland.

**Visualization:** Ipsita Das.

**Writing – original draft:** Ipsita Das, Marc Jeuland.

**Writing – review & editing:** Ipsita Das, Jessica J. Lewis, Ramona Ludolph, Melanie Bertram, Heather Adair-Rohani, Marc Jeuland.

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
