## [Decision Letter · Decision Letter 0]

19 Oct 2020

PONE-D-20-17182

The benefits of action to reduce household air pollution (BAR-HAP) model: A new decision support tool for energy and health sector decision-makers

PLOS ONE

Dear Dr. Jeuland,

Thank you for submitting your manuscript to PLOS ONE. After careful consideration, we feel that it has merit but does not fully meet PLOS ONE’s publication criteria as it currently stands. Therefore, we invite you to submit a revised version of the manuscript that addresses the points raised during the review process.

We look forward to receiving your revised manuscript.

Kind regards,

Fausto Cavallaro, PhD

Academic Editor

PLOS ONE

2.We note that you have stated that you will provide repository information for your data at acceptance. Should your manuscript be accepted for publication, we will hold it until you provide the relevant accession numbers or DOIs necessary to access your data. If you wish to make changes to your Data Availability statement, please describe these changes in your cover letter and we will update your Data Availability statement to reflect the information you provide.

**Comments to the Author**

1. Is the manuscript technically sound, and do the data support the conclusions?

Reviewer #1: Partly

Reviewer #2: Yes

Reviewer #3: Partly

Reviewer #4: Yes

2. Has the statistical analysis been performed appropriately and rigorously? 

Reviewer #1: N/A

Reviewer #2: Yes

Reviewer #3: Yes

Reviewer #4: N/A

3. Have the authors made all data underlying the findings in their manuscript fully available?

Reviewer #1: No

Reviewer #2: Yes

Reviewer #3: Yes

Reviewer #4: Yes

4. Is the manuscript presented in an intelligible fashion and written in standard English?

Reviewer #1: Yes

Reviewer #2: Yes

Reviewer #3: Yes

Reviewer #4: Yes

5. Review Comments to the Author

Reviewer #1: This is a paper that presents a new decision-support tool for assessing the impacts of cleaner cooking fuel on multiple sustainability-relevant goals, including household air pollution. It builds on a prior cost-benefit model published in 2018, which was expanded in this work. The tool was piloted in a case study in Nepal.

While the tool seems appropriate for the problem and builds strongly on other peer-reviewed work, several revisions could make this paper clearer and more well-supported.

1) It is unclear which household air pollutants are accounted for as influencing health, and which as influencing climate. For example, on page 1, black carbon is noted only as a climate forcing agent — is its influence on health also calculated? What about other household air pollutants?

2) This raises a broader question of the detailed equations, which are missing from the paper. It is stated they will be made available, but it is nearly impossible to do a thorough technical review of this manuscript without this detailed information to benchmark the realism of the simulations.

3) While the tool’s specific domain is Nepal, it is noted that it could be applied elsewhere, and that data will need to be updated. Some more detailed discussion about model limitations (other than data) in applicability to other regions would be useful. To what extent do the fundamental assumptions about processes simulated hold across the world?

4) In the case presented of Nepal, it is unclear whether this is designed as an experimental setting or not, and whether the paper intends to present results from this experiment. If this is the case, more formality in data collection and what questions were tested in this experiment would be useful.

5) Relatedly, the paper does not seem to ask and answer a relevant scientific question related to the tool, so it’s unclear what exactly the scientific contribution of this work is, and to which field. Is this an advance on the decision-support tool itself, or the methodology? This could be made much clearer in a revision.

Reviewer #2: The paper deals an interesting issue, i.e. cooking with polluting and inefficient fuels and technologies, very interesting especially in the investigated area, Nepal. The authors presented a new model, BAR-HAP, for evaluating and considering the health, environment and development implications in such topic. The approach followed by authors is good, really persuasive, well-reported and presented, following all the research and publication ethics. I suggest to stress or show a comparison with different situations worldwide, to deep the effects on the air quality, to highlight the possibility to export this approach in other situations, to expand the conclusions (they are very poor, 5 lines!).

Reviewer #3: Comments for Manuscript titled: The benefits of action to reduce household air pollution (BAR-HAP) model: A new decision support tool for energy and health sector decision-makers

The manuscript is an illustrative work to showcase the functional characteristics, usability and limitations of the existing Household Air Pollution (BAR-HAP) Tool developed by WHO. The work can be informative and helpful to someone who wants to explore the model in their respective region of study. The model has been explained elaborately with a pertinent case study of Nepal. However, there are some minor corrections before it can be accepted for publication, mostly the readability of the manuscript. There is also a need to revise the textual flow as the organic linkage of statements to each other is missing at some places. Please go through the following comments and address them carefully.

1. Section 1 Introduction: Please correct typographical errors and add missing information in the manuscript. A few instances listed below-

Page 1 “eliminate the use unprocessed coal…” .

Page 5 Conceptual framework –Please expand and define “DALYs”

2. Please provide information of the study area and its important climatic and geographical characteristics

3. Information regarding types of biomass fuels used in the study area is missing

4. Section 3 Methods - is too lengthy and requires sub sections to improve readability. Please add sub headings such as Policy instruments, Model assumptions etc.

5. A separate section is needed to give brief information about the clean cooking interventions and performance metrics

6. The results section should be clearly demarcated as done for other sections

7. What about model validation? Is there any option for that? If not, it should be clearly mentioned in the paper.

8. “Despite this, total social net benefits are only slightly reduced relative to the LPG only transition, because the natural draft biomass stove still provides significant fuel and time savings, and climate mitigation, previously shown to be important in Figure 3 Benefits can also be generated at somewhat lower subsidy levels, owing to the lower cost of the natural draft stoves.”

Observation regarding Climate mitigation potential of natural draft stoves is not correct. Studies have shown that these stoves BC even higher than traditional cookstoves.

9. “Furthermore, users should understand the limitations of the tool, discussed in this article, including the fact that it only covers the contribution of cooking practices to HAP (and does not cover lighting and heating, for example).”

There are a few other limitations that should be admitted in this paper:

i) The exposures vary widely depending on the structural characteristics of household, type of solid biomass fuels used and user behaviour which I am not sure has been incorporated in the model. These variations may induce appreciable changes in the overall exposure of the user to the emissions and eventually the health outcomes. This should be added to limitations.

ii) Indoor air quality concentrations are also impacted by ambient air quality of the immediate locality which can sometimes nullify the reductions brought about by the use of clean cooking in devices. The immediate air pollutant levels high also due to partial uptake of clean cookstoves in a village for example, while most the households are using traditional cookstoves which allows emissions to get released into the outside environment. This also been ignored in the model.

10. The modifications done in the model with respect to area of study should be clearly demarcated and explained so that it is clear to the reader how much the results can vary through such modifications.

11. Quality of the figures can be improved with better resolution.

Reviewer #4: The benefits of action to reduce household air pollution (BAR-HAP) model: A new

decision support tool for energy and health sector decision-makers

The authors present a novel decision tool, to help policy makers related to energy, be able to project the effect of different policies on health outcomes along with the cost benefit analysis. There are not many decision tools available in resource limited setting to help decide on the policy selection. Though similar framework had been used to evaluate effects of intervention in Bangladesh. Some suggestions

1. Title could be shortened

2. Introduction could be shortened

3. In methods, subsections could be added to conceptual framework since the description is lengthy. And difficult to follow the flow of the manuscript. Subsections could include: Policy interventions available, possible transitions, costs and benefits of intervention, cost-benefit analysis, process of actually performing the analysis, advantages and limitations.

4. In discussion, comparing BAR-HAP tool with other similar energy policy-based tool would help. [e.g. (LEAP-IBC) tool to assess air quality and climate co-benefits, GridPIQ/Combined Heat and Power (CHP) Energy and Emissions Savings Calculator] that help to assess the effect of energy transition.

5. In figure 2 if available, to include what the predicted uptake for each intervention in year 1,2, 3…., or it could be included in a Table

6. In page 14, the paragraph before Figure 3, ‘The cost effectiveness ratios (CER) range between about 674 and 869 US$/DALY avoided across subsidy levels’. Is there any data when the intervention would start becoming benefit (positive) independent of the DALY avoided? That is, when could we expect the net social benefit to equalize the cost invested, at 70% uptake.

7. In Figure 4, the colors are very similar. If possible, to change to more distinct colors.

6. PLOS authors have the option to publish the peer review history of their article (what does this mean?). If published, this will include your full peer review and any attached files.

Reviewer #1: No

Reviewer #2: No

Reviewer #3: No

Reviewer #4: No

---

## [Author Response · Author response to Decision Letter 0]

27 Nov 2020

Please see our attached point by point response to the reviewer and editor comments.

---

## [Decision Letter · Decision Letter 1]

7 Jan 2021

The benefits of action to reduce household air pollution (BAR-HAP) model: A new decision support tool

PONE-D-20-17182R1

Dear Dr. Jeuland,

We’re pleased to inform you that your manuscript has been judged scientifically suitable for publication and will be formally accepted for publication once it meets all outstanding technical requirements.

Kind regards,

Fausto Cavallaro, PhD

Academic Editor

PLOS ONE

Additional Editor Comments (optional):

Reviewers' comments:

Reviewer's Responses to Questions

**Comments to the Author**

1. If the authors have adequately addressed your comments raised in a previous round of review and you feel that this manuscript is now acceptable for publication, you may indicate that here to bypass the “Comments to the Author” section, enter your conflict of interest statement in the “Confidential to Editor” section, and submit your "Accept" recommendation.

Reviewer #1: All comments have been addressed

Reviewer #2: All comments have been addressed

Reviewer #3: All comments have been addressed

Reviewer #4: All comments have been addressed

2. Is the manuscript technically sound, and do the data support the conclusions?

Reviewer #1: Yes

Reviewer #2: Yes

Reviewer #3: Yes

Reviewer #4: Yes

3. Has the statistical analysis been performed appropriately and rigorously? 

Reviewer #1: N/A

Reviewer #2: N/A

Reviewer #3: Yes

Reviewer #4: Yes

4. Have the authors made all data underlying the findings in their manuscript fully available?

Reviewer #1: Yes

Reviewer #2: Yes

Reviewer #3: Yes

Reviewer #4: Yes

5. Is the manuscript presented in an intelligible fashion and written in standard English?

Reviewer #1: Yes

Reviewer #2: Yes

Reviewer #3: Yes

Reviewer #4: Yes

6. Review Comments to the Author

Reviewer #1: All comments have been well addressed in this revision. The paper is clearer in its focus and its presentation is improved.

Reviewer #2: The authors made a good effort for responding to the referees' questions and they increased the paper quality. Now the paper could be considered for publication in Plo One.

Reviewer #3: All the comments have been satisfactorily addressed by the authors. The manuscript now looks more organized with a better contextual flow.

Reviewer #4: Thanks for the revised manuscript. The manuscript has a better flow and more user friendly terms to understand the tool (with its benefits and limitations).

7. PLOS authors have the option to publish the peer review history of their article (what does this mean?). If published, this will include your full peer review and any attached files.

Reviewer #1: No

Reviewer #2: No

Reviewer #3: No

Reviewer #4: No

---

## [Editor Report · Acceptance letter]

8 Jan 2021

PONE-D-20-17182R1 

The benefits of action to reduce household air pollution (BAR-HAP) model: A new decision support tool 

Dear Dr. Jeuland:

I'm pleased to inform you that your manuscript has been deemed suitable for publication in PLOS ONE. Congratulations! Your manuscript is now with our production department. 

Kind regards, 

on behalf of

Professor Fausto Cavallaro 

Academic Editor

PLOS ONE